# TKG-LM: Temporal Knowledge Graph Extrapolation Enhanced by Language Models

## Abstract

Temporal Knowledge Graph (TKG) extrapolation aims to predict future missing facts based on historical information. While graph embedding methods based on TKG topology structure have achieved satisfactory performance, the semantic text information of entities and relations still needs to be fully exploited. As large language models (LMs) such as ChatGPT sweep the entire field of natural language processing field, considerable works about KGs augment LMs with structured representations of world knowledge. In this paper, we proposed a method called TKG-LM to fill the gap in the effective integration of TKG and LMs, including historical events pruning, sampling prompt construction, and layer-wise modality fusion. Specifically, we adopt a pruning strategy to extract valuable events from numerous historical facts and reduce the search space for answers. Then, LMs and time-weighted functions are adopted to score the semantic similarity of each neighbor tuple, and the history-sampling prompt is built as the input of LMs. We integrate the encoded representation of LMs and graph neural networks in a multi-layer framework to enable bidirectional information flow between the modalities. This facilitates the incorporation of structured topology knowledge into the language context representation while leveraging linguistic nuances to enhance the graphical representation of knowledge. Our TKG-LM outperforms state-of-the-art (SOTA) TKG methods on five standard TKG datasets and beats the existing LLM and LM+KG models. Further ablation experiments demonstrate the role of our module designs and the benefits of integrating LM and GNN representation.

## 1 Introduction

Temporal knowledge graph (TKG) (Ji et al., 2021; Wang et al., 2023) , as a universal format for describing facts in the real world, can record the relationships between entities and the timestamps when these relationships are established. Most TKG data is typically represented as quadruples (subject, relation, object, and time) and can serve as the perfect knowledge base for answering questions about when certain events occur. For example, (Barack Obama, Make a visit, France, February 10) indicates that Obama made a visit to France on February 10. In this paper, we mainly address the extrapolation over TKGs, which requires modeling the dynamics of events along the timeline and forecasting future incomplete facts (including missing entities and relations) based on historical information. Extrapolation tasks have attracted widespread attention due to their enormous practical significance and are gradually playing an important role in many natural language processing (NLP) tasks, such as open-world knowledge completion (Shi & Weninger, 2018), crisis event forewarning (Luo et al., 2020), and financial forecasting (Cheng et al., 2022).

Due to the abundant dynamics of facts on TKG, previous works towards TKG extrapolation (Seo et al., 2018; Jin et al., 2019; Zhu et al., 2021) mainly focus on how to mine the most valuable historical events associated with future facts based on the connected graph topology of TKG. In general, these approaches model structured knowledge with the evolutionary interactions of a series of graph snapshots, and predict the score of new facts based on hidden representations in the embedding space. Despite their simplicity and effectiveness, the sufficient structured semantics of TKG can hardly be represented by semantic-agnostic GNNs, which largely wastes the powerful reasoning potentials of language nuances. Recently, pre-trained Language Models (LMs) have advanced the capability of natural language understanding and reasoning by a remarkable progress from BERT (Devlin et al., 2018) to GPT-3 (Brown et al., 2020). Driven by the noteworthy success

of language models (LMs), extensive efforts have been directed towards integrating pre-trained LMs with diverse domains, such as dynamic graphs (Zhang, 2023), knowledge graphs (Yao et al., 2019; Yasunaga et al., 2021), and tables (Jiang et al., 2023). Regarding the most relevant contributions to TKG, KG-BERT (Yao et al., 2019)incorporates structured knowledge from KGs into both pretraining and fine-tuning processes to enhance semantic modeling of KGs. Nevertheless, the dynamic nature of TKG complicates the application of these aforementioned methods designed for KGs to the extrapolation of TKG.

To the best of our knowledge, the integration of expressive LMs with structured TKGs still remains unexplored, which mainly faces the following three challenges: **(i) Adequate utilization of the semantic prior knowledge of LMs.** Existing works usually directly utilize historical events structurally related to the entity (such as static graphs (Li et al., 2021) and global histories (Li et al., 2022)), but these events contain considerable context-free factors of the prediction. We can effectively capture valuable events from numerous historical facts by using LMs. **(ii) Robust temporal reasoning**. LMs trained on static corpora present a very serious defect in processing temporal extrapolation tasks. Manually constructing topology-relevant prompt instructions will cause LMs to over-rely on simple, or even spurious, patterns to find shortcuts to answers, leading to overfitting and reducing generalization (Wang et al., 2021; Lee et al., 2023). **(iii) Effectively interaction of multimodal information.** How to effectively fuse the two modalities of graph embedding and linguistic representation for joint reasoning is an import and opening qusetion. The modality interaction is limited in scope and degree and is usually performed in a shallow or non-interactive manner (Either there is no interaction or one kind of information is used to augment the other) (Yasunaga et al., 2021; Zhang et al., 2022). In addition, LMs has a weak perception of spatial and topological factors, and they are unable to perform precise multi-step computational reasoning on history snapshots as GNNs can. (Lee et al., 2023)

To overcome the above three challenges and benefit from graph embedding and text encoding advantages, we propose a model that enhances the TLG extrapolation by LMs (called TKG-LM), mutually beneficial to learning contextual semantic information and topologically structured knowledge. Specifically, we propose a time-weighted LM-based function to score the gap between the query to be predicted and its historical fact. The most relevant events are retained and retrieved as the subgraph for subsequent inference. To alleviate the pain point of LM's poor performance on temporal reasoning tasks, we devise an adaptive prompt based on sampling as the input of LMs. We then feed the pruned subgraph and the above prompt into the GNN and LM layer, respectively. The layer-wise modality interaction is composed of an attention-based residual fusion module. It can effectively integrate the explicit information provided by TKG semantic context with the implicit association between knowledge to carry out powerfully structured reasoning. The optimization objective involves the loss of predicting entity and relation, as well as the loss of reconstructing target tokens by the LM.

We evaluate the performance of our TKG-LM on five commonly used TKGs: the ICEWS series dataset in the crisis warning domain (Trivedi et al., 2017), the global event dataset GDELT (Leetaru & Schrodt, 2013), and the dataset YAGO (Mahdisoltani et al., 2013) about Wikipedia and knowledge. TKG-LM outperforms methods from four different domains: existing state-of-the-art (SOTA) TKG embedding baselines, fine-tuned LM, Graph+LM, and KG+LM models. Further ablation experiments demonstrate the role of our module designs and the benefits of integrating LM and GNN representation.

## 2 RELATED WORK

**TKG Extrapolation Learning**    In this section, we review the existing methods for TKG extrapolation tasks. TTransE (Leblay & Chekol, 2018) incorporates temporal constraints and encodes this information into a translation similar to RNN relations. CyGNet (Zhu et al., 2021) proposes a copy-generation mechanism that uses repeated patterns in historical facts to predict future facts while ignoring higher-order semantic dependencies between concurrent entities. RE-NET (Li et al., 2021) models the long-term relationship of the entities to be predicted as a sequence and combines RNN to replenish temporal and structural dependencies. xERTE (Han et al., 2020) designs a representation update mechanism that iteratively propagates attention along sampled edges to mimic human reasoning behavior. RE-GCN (Li et al., 2021) focuses on evolutionary dynamics in TKG and generates

entity embeddings by modeling a sequence of historical graph snapshots. TiRGN (Li et al., 2022) proposes a local-global history pattern time-guided recurrent graph network to consider different laws of historical facts comprehensively. The above methods ignore the large amount of textual information inherent in historical facts of TKGs and do not absorb the advantages of multi-modal KG reasoning models.

**LM-Enhanced Models** In recent years, Language Models (LMs) especially GPT-3 (Brown et al., 2020), have achieved impressive performance on natural language processing tasks. Their extensions are gradually used to solve problems with data from different modalities. KG-Bert (Yao et al., 2019) is the first to utilize a pre-trained language model for knowledge graph reasoning tasks. Kg-S2S (Cheng et al., 2022) will unify the representation of facts in KG as flat texts to handle different linguistic graph structures. Graph-BERT (Zhang et al., 2020) integrates graph structure into BERT-style models, allowing them to learn graph representations effectively. It leverages self-attention mechanisms for learning on both textual and graph data. GraphToolFormer (Zhang, 2023) hand-craftes instructions and a few hint templates for graph reasoning tasks and allowes LMs to call appropriate external API functions to augment the dataset of reasoning statements. StructGPT (Jiang et al., 2023) utilizes external interfaces to accurately access and filter structured data and further iterates this step using LM's reasoning capabilities. However, the modality interaction of the above works is limited in scope and degree and is usually performed in a shallow or non-interactive manner (either there is no interaction or one kind of information is used to augment the other). In the paper, we explore deeper integrations of both topological and semantic modality.

## 3 PROPOSED METHOD: TKG-LM

In this section, we discuss new approaches to incorporating textual data into temporal knowledge graph embeddings. As for the extrapolation over TKGs, we detail our TKG-LM from the following four aspects: 1) **Scoring and Pruning (§3.1).** To handle the vast amount of historical facts in TKG, we utilize the language model's prior knowledge to filter out irrelevant events, thereby decreasing the search space for answers. The relevance measure is a scoring function that combines semantic and temporal information. 2) **Sampling Prompt Construction (§3.2).** To improve the robustness of LMs on TKG extrapolation prediction, we construct adaptive and variable sampling prompt instructions as the input of LMs. 3) **Layer-wise Modality Fusion §3.3.** Our method integrates the encoded representation of LMs and graph neural networks in a multi-layer attention-based module to enable bidirectional information flow between two modalities, while leveraging linguistic nuances to enhance the graphical representation of knowledge. Finally, the embedding of entities and relations learned by the model are decoded into probabilities, and the optimization objective is calculated.

**Definitions** A TKG is formalized as a directed multi-relational dynamic graph $\mathcal{G} = \{\mathcal{E}, \mathcal{R}, \mathcal{T}, \mathcal{F}\}$ with a set of entities $\mathcal{E}$, relations $\mathcal{R}$, timestamps $\mathcal{T}$, and facts (edges) $\mathcal{F}$. Each fact $f \in \mathcal{F}$ that occurs at time $t$ is described as a quadruple $(s, r, o, t)$, where $o, s \in \mathcal{E}$ are head and tail entities respectively while $r \in \mathcal{R}$ is their relation. The inverse quadruple $(o, r^{-1}, s, t)$ will be appended to $\mathcal{F}$, and the TKG $\mathcal{G}$ is split into a sequence of time-stamped snapshots $\mathcal{G} = \{\mathcal{G}_1, ..., \mathcal{G}_T\}$. The TKG extrapolation aims to forecast a future missing object, subject, or relation according to previous historical subgraphs. Formally, given queries $(s, r, ?, t), (?, r, o, t)$ and $(s, ?, o, t) \in \mathcal{G}_t$, we learn a function $f(\cdot)$ that predicts the conditional probability $p$ of all entities and relations:

$$\mathcal{G}_{1:t-1} = \{\mathcal{G}_1, ..., \mathcal{G}_{t-1}\} \xrightarrow{f(\cdot)} p(o \mid s, r, t), p(s \mid o, r, t), \text{and } p(r \mid s, o, t). \tag{1}$$

### 3.1 SCORING AND PRUNING

Given a query $q = (s, r, ?, t) \in \mathcal{G}_t$ to be predicted, let $N_{(m)}^s$ be all the $m$-hop neighbors of the entity $s$ which are retrieved from the historical snapshots $\mathcal{G}_{1:t-1}$. We define the $m$-hop subgraph of $q$ as the induced subgraph of $s \cup N_{(m)}^s$, *i.e.* $\mathcal{G}_{(m)}^q = \mathcal{G}_{1:t-1}[s \cup N_{(m)}^s]$, then the fact set of $\mathcal{G}_{(m)}^q$ is:

$$\mathcal{F}_{(m)}^q = \{f : (u, r_f, v, t_f) \mid u, v \in s \cup N_{(m)}^s, t_f < t, f \in \mathcal{F}_{1:t-1} \subset \mathcal{G}_{1:t-1}\}, \tag{2}$$

where $f$ is a fact quadruple, $\mathcal{F}_{1:t-1}$ is the fact set of $\mathcal{G}_{1:t-1}$, and $\mathcal{F}_{(m)}^q$ represents the set of historical events topologically related to $q$. Existing works usually directly utilize $\mathcal{F}_{(m)}^q$ for representation

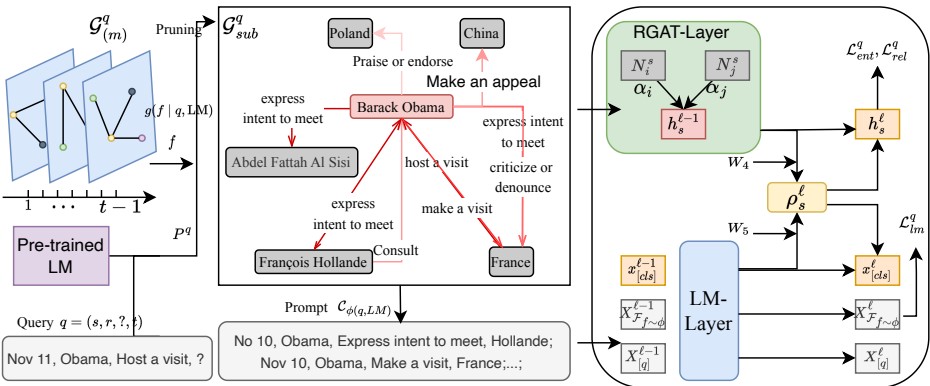

Figure 1: Framework of our TKG-LM. The fact $f$ in the m-hop subgraph $\mathcal{G}^q_{(m)}$ of the query q is scored by a pre-trained LM and pruned to $G^q_{sub}$. The darker the red edge, the more relevant $q$ and the historical facts are, and the higher the probability of being sampled as the prompt $\mathcal{C}_{\phi(q,\text{LM})}$. The two modality information represented by RGAT and LM-Layer will interact layer-by-layer by a residual attention module. The optimization object includes the loss for entity prediction $\mathcal{L}^q_{ent}$, relation prediction $\mathcal{L}^q_{rel}$, and target token reconstruction $\mathcal{L}^q_{lm}$.

learning, such as building the static graph of snapshots (Li et al., 2021) or considering global histories of entities (Li et al., 2022)). However, $\mathcal{F}^q_{(m)}$ contains considerable interference events irrelevant to the query. Confounding multiple related and irrelevant factors together without effective extraction may lead to overfitting and introduce uncertainty.

Considering the above limitations, we innovatively propose a time-aware pruning strategy based on LMs, which can extract valuable historical events from $\mathcal{F}^q_m$. It is known that LMs learn to implicitly encode extensive knowledge about the world after being trained on a common text anticipation library. To make fully use of the semantic prior knowledge of LMs and accurately infer future facts, our strategy adopts a pre-trained LM such as BERT or RoBERTa to score the semantic gap between each historical fact and the query. In a manner akin to the training method used for masked language models (MLMs), we treat historical facts as the masked text and the query $q$ as their surrounding text. Formally, the probability vector $P^q$ of all tokens in the vocabulary is predicted by $q$:

$$P^q = \text{Head}\left(\text{LM-Encoder}\left(\text{Tokenizer}\left([\text{Text}(t); \text{Text}(s); \text{Text}(r)]\right)\right)\right) \in \mathbb{R}^{\text{vocab-size}}, \quad (3)$$

where LM-Encoder$(\cdot)$ denotes the LM encoder consisting of multiple transformers, producing contextualized representation for the sentence $[\text{Text}(t); \text{Text}(s); \text{Text}(r)]$. Head$(\cdot)$ is an MLP classification layer that takes the hidden representation from the LM encoder to predict each token's probability, and vocab-size is the size of the vocabulary.

For each historical fact $f = (u, r_f, v, t_f) \in \mathcal{F}^q_{(m)}$, we concatenate the text $[\text{Text}(u); \text{Text}(r_f); \text{Text}(v)]$ and tokenize the combined sequence into masked token indices $X = \{x_1, ..., x_L\}$. The time-weighted semantic gap between $f$ and $q$ is calculated by:

$$g(f \mid q, \text{LM}) = (1 - r_i) \cdot \mathcal{L}_{\text{MLM}} = (1 - \exp\left(-(t - t_f)\right)) \cdot \frac{-1}{|X|} \sum_{x \in X} \log P^q_x, \quad (4)$$

where $r_i$ is the time-weighted coefficient, which exponentially increases in the negative direction as the time difference increases. $\mathcal{L}_{\text{MLM}}$ is the cross-entropy loss for masked token reconstruction, and its value measures the semantic gap between each historical fact and the query. The larger $g(f \mid q, LM)$ is, the larger is the gap between $f$ and $q$, so we pick $n$ historical facts with the smallest $g(f \mid q, LM)$ to build a pruned subgraph of $\mathcal{G}^q_{(m)}$ called $\mathcal{G}^q_{sub}$.

The process of pruning irrelevant historical facts aims to decrease candidate entities and relations. This improves the training process's efficiency and enhances prediction performance, which is verified in the ablation experiments. For example, considering the 2-hop subgraph of the quadruple (Barack Obama, Host a visit, François Hollande, Feb 10) in the ICEWS14 dataset, its number of historical events is close to 3000. As shown in Figure 1, "*China*" and "*Poland*" are irrelevant entities, which may introduce unnecessary difficulties in inference.

## 3.2 SAMPLING PROMPT CONSTRUCTION

There are two inherent drawbacks to LMs: (i) have a weak perception of spatial and topological structure, and (ii) unable to perform accurate multi-step computational reasoning. To alleviate the above drawbacks, existing works manually construct topology-relevant instructions to fine-tune LMs (Zhang, 2023; Jiang et al., 2023). However, when it comes to the temporal extrapolation tasks, directly using the fixed prompt template will present a very serious defect. It causes LMs to overrely on simple, or even spurious, patterns to find shortcuts to answers, leading to overfitting and reducing generalization (Wang et al., 2021; Lee et al., 2023).

In order for LMs to learn a robust representation of the underlying relationships between facts, we construct sampling prompts based on the prior scoring function of §3.1. Specifically, we normalize the gap $g(f \mid q, LM)$ of each fact $f$ on the pruned subgraph $\mathcal{G}_{sub}^q$ by the softmax function:

$$\Phi(q, \text{LM}) : \varphi(f \mid q, \text{LM}) \triangleq 1 - \operatorname*{softmax}_{f \in \mathcal{F}_{sub}^q} \{g(f \mid q, \text{LM})\}, \tag{5}$$

where $\mathcal{F}_{sub}^q$ is the fact set of $\mathcal{G}_{sub}^q$, $\varphi(f \mid q, LM)$ is the relevent probability between $f$ and $q$, and $\Phi(q, LM)$ is the corresponding probability distribution. During each training iteration, $k$ historical events in $\mathcal{F}_{sub}^q$ are sampled according to $\Phi(q, LM)$:

$$\mathcal{F}_{f \sim \Phi(q, \text{LM})} := \{f_i \mid f_i \sim \Phi(q, \text{LM}), f_i \in \mathcal{F}_{sub}^q \subset \mathcal{G}_{sub}^q\}_{i=1}^k. \tag{6}$$

Then the context of the sampling prompt of $q = (s, r, ?, t)$ is buid as:

$$\mathcal{C}_{\Phi(q, \text{LM})} = \text{Text}\left([\mathcal{F}_{f \sim \Phi(q, \text{LM})}; q]\right) = \text{Text}\left([t_{f_1}, u_1, r_{f_1}, v_1; \dots; t_{f_k}, u_k, r_{f_k}, v_k; t, s, r, ?]\right). \tag{7}$$

Our method strives to answer time-sensitive questions, explore knowledge graphs, infer time information from questions, and help models find answers.

## 3.3 LAYER-WISE MODALITY FUSION

An important and opening question is how to effectively fuse the two modalities of graph embedding and linguistic representation for joint reasoning. The modality interaction of previous works is limited in scope and degree and is usually performed in a shallow or non-interactive manner (either there is no interaction or one kind of information is used to augment the other) (Yasunaga et al., 2021; Zhang et al., 2022). On the one hand, LMs has a weak perception of spatial and topological factors, and they are unable to perform precise multi-step computational reasoning on history snapshots as GNNs can. On the other hand, LMs can enhance the ability to capture valuable free text. It also augments the text encoding paradigm by modeling contextual knowledge, which is crucial for graph-related tasks.

To enhance the mutual interactions between text and TKGs, we utilize a residual attention-based module to fuse modalities. This process involves taking in text tokens and TKG node embeddings and exchanging information between them by layer-wise encoders to create a fused representation for each token and the query. Specifically, the sampling prompt $\mathcal{C}_{\Phi_{q, \text{LM}}}$ is mapped into initial token embeddings $[x_{[cls]}^0; X_{[\mathcal{F}_{f \sim \Phi(q, \text{LM})}]}^0; X_{[q]}^0]$ by an LM-Embedding layer:

$$[x_{[cls]}^0; X_{[\mathcal{F}_{f \sim \Phi(q, \text{LM})}]}^0; X_{[q]}^0] = \text{LM-Embedding}\left(\text{Tokenizer}\left(\mathcal{C}_{\Phi(q, \text{LM})}\right)\right). \tag{8}$$

And $N$ layers of the LM are adopted to update the textual modality representations:

$$[\widetilde{x}_{[\text{cls}]}^\ell; X_{[\mathcal{F}_{f \sim \Phi(q, \text{LM})}]}^\ell; X_{[q]}^\ell] = \text{LM-Layer}^\ell\left([x_{[\text{cls}]}^{\ell-1}; X_{[\mathcal{F}_{f \sim \Phi(q, \text{LM})}]}^{\ell-1}; X_{[q]}^{\ell-1}]\right), \ell = 1, \cdots, N. \tag{9}$$

Then we extract topological modality representations on $q$'s pruned graph $\mathcal{G}_{sub}^q$ by performing the attention mechanism (Vaswani et al., 2017). The $l$-th layer attention coefficient $\alpha_{u,r,v,t}^\ell$ between each subject $u \in \mathcal{G}_{sub}^q$ and its triple neighbor tuples $(r, v, t) \in N^u$ is denoted as:

$$\alpha_{u,r,v,t}^\ell = \operatorname*{softmax}_{(r,v,t) \in N^u \subset \mathcal{G}_{sub}^q} \{\text{ReLU}\left(a^T W_1^\ell [h_u^{\ell-1} \| h_v^{\ell-1} \| z_r \| \psi(t^+ - t)]\right)\}, \tag{10}$$

$$\psi(\Delta t) = [\cos(w_1 \Delta t + b_1), \dots, \cos(w_d \Delta t + b_d)] \in \mathbb{R}^d, \tag{11}$$

where ReLU is the activation function, $T$ is the transposition, and $\|$ is the concatenation. $a \in \mathbb{R}^{3d}$ and $W_1^{\ell} \in \mathbb{R}^{3d \times 3d}$ are learnable parameters. $\psi(\Delta t)$ is a time encoding function and $t^+$ is the maximum over all times in $N^u$. $z_r \in \mathbb{R}^d$ is the embedding of relation $r$, and $h_u^{\ell} \in \mathbb{R}^d$ is the $l$-th layer embedding of $u$, which is updated by the previous layer:

$$\widetilde{h}_u^{\ell} = \sum_{(r,v,t) \in N^u \subset \mathcal{G}_{sub}^q} \alpha_{u,r,v,t}^{\ell} W_2^{\ell}(h_v^{\ell-1} + z_r) + W_3^{\ell} h_u^{\ell-1}, \ell = 1, \ldots, M, \tag{12}$$

where $W_2^{\ell}$ and $W_3^{\ell}$ are learnable parameters, $M$ is the number of GNN layers. As for the LM-GNN fusion branch, When the layer number is N-M to N: The first $N$ layers of LM extract hidden representations of the prompt, the representation corresponding to the first token $[cls]$ is fused with the node embeddings of GNN, and after concatenating, an attention-based module is used for fusion:

$$\rho_s^{\ell} = \text{softmax}\{\tanh(W_4 \widetilde{x}_s^{\ell}) a_3, \tanh(W_5 \widetilde{h}_s^{\ell}) a_2\} \in \mathbb{R}^2 \tag{13}$$

$$h_s^{\ell} = \sigma\left(\rho_{s0}^{\ell} W_4 \widetilde{x}_{[cls]}^{\ell} + \rho_{s1}^{\ell} W_5 \widetilde{h}_s^{\ell}\right) + \widetilde{h}_s^{\ell} \tag{14}$$

$$x_{[cls]}^{\ell} = \sigma\left(\rho_{s0}^{\ell} W_4 \widetilde{x}_{[cls]}^{\ell} + \rho_{s1}^{\ell} W_5 \widetilde{h}_s^{\ell}\right) + \widetilde{x}_{[cls]}^{\ell} \tag{15}$$

where $W_4 \in \mathbb{R}^{d \times d}$ and $W_5 \in \mathbb{R}^{d \times d}$ are two learnable matrices for translating the original representations of two modalities into the same embedding space. $\rho$ is the adaptive fusion ratio for representations, which is normalized by the softmax function. Residual connections are introduced so that deeper modality representations can learn additional information from the original features obtained from earlier layers. It facilitates capturing more abstract and complex pattern interactions, enabling our layer-wise fusion to create richer representations.

Let $o$ is the target object of $q$, then the language model's loss $\mathcal{L}_{lm}^q$ is defined as the probability of predicting $o$'s text Text($o$) using the $N$-th layer contextual representation $[x_{[cls]}^N; X_{[\mathcal{F}_{f \sim \Phi(q,\text{LM})}]}^N; X_{[q]}^N]$:

$$\mathcal{L}_{lm}^q = -\frac{1}{|W|} \sum_{w \in W} \log p\left(w_i \mid [x_{[cls]}^N; X_{[\mathcal{F}_{f \sim \Phi(q,\text{LM})}]}^N; X_{[q]}^N]\right), \tag{16}$$

where $W = \{w_1, \cdots, w_L\}$ is the tokens of Text($o$). The intuition is that $\mathcal{L}_{lm}^q$ enables the model to jointly use structured knowledge in the text and TKG to reason about masked tokens in the text.

### 3.4 OBJECT OPTIMIZATION

We utilize ConvTransE (Li et al., 2021; Shang et al., 2019) as a decoder to predict probabilities of entity prediction and relation prediction task. Their loss is calculated by Cross-Entropy:

$$\mathcal{L}_{ent}^q = -\log p\left(o \mid s, r, t, \mathcal{G}_{1:t-1}\right) = -\log \sigma\left(\text{MLP}\left(\text{Conv}([\mathbf{h}_s \| \mathbf{z}_r \| \psi_t])\right) \cdot \mathbf{h}_o\right), \tag{17}$$

$$\mathcal{L}_{rel}^q = -\log p(r \mid s, o, t, \mathcal{G}_{1:t-1}) = -\log \sigma\left(\text{MLP}\left(\text{Conv}([\mathbf{h}_s \| \mathbf{h}_o \| \psi_t])\right) \cdot \mathbf{z}_r\right), \tag{18}$$

where MLP is a multi-layer perceptron and Conv is a convolution. $\mathbf{h} \in \mathbb{R}^{|\mathcal{E}| \times d}$ and $\mathbf{z} \in \mathbb{R}^{|\mathbb{R}| \times d}$ are the embedding of all entities and relations after the fusion of LM and GNN layers. The optimization objective $\mathcal{L}$ is the sum of $\mathcal{L}_{ent}^q, \mathcal{L}_{ent}^q$ and $\mathcal{L}_{ent}^q$ for each query $q$ on the snapshot $\mathcal{G}_t$ at each time $t$:

$$\mathcal{L} = \sum_{t=1}^{T} \sum_{q \in \mathcal{G}_t} \alpha \cdot \mathcal{L}_{ent}^q + (1 - \alpha) \cdot \mathcal{L}_{rel}^q + \beta \cdot \mathcal{L}_{lm}^q, \tag{19}$$

where $\alpha$ and $\beta$ are two hyper-parameters to represent the weight of different tasks.

## 4 EXPERIMENT

### 4.1 EXPERIMENTAL SETUP

**Dataset.** Five public TKG datasets are adopted to verify the effectiveness of our proposed TKG-LM, including GDELT (Leetaru & Schrodt, 2013), ICEWS14 (Trivedi et al., 2017), ICEWS05-15 (García-Durán et al., 2018), ICEWS18 (Boschee et al., 2015) and YAGO (Mahdisoltani et al., 2013). Following extensive previous work (Li et al., 2021; 2022), we split the datasets into training, validation, and testing set with ratios 8:1:1 chronologically. The detailed information about the involved datasets is in Appendix 5.

Table 1: MRR (%) and Hit rate (%) (H for short) comparison of TKG embedding methods and our TKG-LM on five commonly used datasets for the TKG extrapolation task. The best results are in **bold**, and the second-best results are underlined.

| Model | GDELT | | | | ICEWS14 | | | | ICEWS05-15 | | | | ICEWS18 | | | | YAGO | | | |
|---|---|---|---|---|---|---|---|---|---|---|---|---|---|---|---|---|---|---|---|---|
| | MRR | H@1 | H@3 | H@10 | MRR | H@1 | H@3 | H@10 | MRR | H@1 | H@3 | H@10 | MRR | H@1 | H@3 | H@10 | MRR | H@1 | H@3 | H@10 |
| DistMult | 8.68 | 5.58 | 9.96 | 17.13 | 20.32 | 6.13 | 27.59 | 46.61 | 19.91 | 5.63 | 27.22 | 47.33 | 13.86 | 5.61 | 15.22 | 31.26 | 44.05 | 25.06 | 49.70 | 61.63 |
| ConvE | 16.55 | 11.02 | 18.88 | 31.60 | 30.30 | 21.30 | 34.42 | 47.89 | 31.40 | 21.56 | 35.70 | 50.96 | 22.81 | 13.63 | 25.83 | 41.43 | 41.22 | 22.27 | 47.03 | 59.90 |
| ComplEx | 16.96 | 11.25 | 19.52 | 32.35 | 22.61 | 9.88 | 28.93 | 47.57 | 20.26 | 6.66 | 26.43 | 47.31 | 15.45 | 8.04 | 17.19 | 30.73 | 44.09 | 24.78 | 49.57 | 59.64 |
| ConvTransE | 16.20 | 10.85 | 18.38 | 30.86 | 31.50 | 22.46 | 34.98 | 50.03 | 30.28 | 20.79 | 33.80 | 49.95 | 23.22 | 14.26 | 26.13 | 41.34 | 46.67 | 26.16 | 52.22 | 62.52 |
| RotatE | 13.45 | 6.95 | 14.09 | 25.99 | 25.71 | 16.41 | 29.01 | 45.16 | 19.01 | 10.42 | 21.35 | 36.92 | 14.53 | 6.47 | 15.78 | 31.86 | 42.08 | 15.68 | 46.77 | 59.39 |
| TTransE | 5.50 | 0.47 | 4.94 | 15.25 | 12.86 | 3.14 | 15.72 | 33.65 | 16.53 | 5.51 | 20.77 | 39.26 | 8.44 | 1.85 | 8.95 | 22.38 | 26.10 | 6.59 | 36.28 | 47.73 |
| RGCRN | 19.37 | 12.24 | 20.57 | 33.32 | 38.48 | 28.52 | 42.85 | 58.10 | 44.56 | 34.16 | 50.06 | 64.51 | 28.02 | 18.62 | 31.59 | 46.44 | 62.76 | 48.25 | 67.56 | 71.69 |
| RE-NET | 19.55 | 12.38 | 20.80 | 34.00 | 39.86 | 30.11 | 44.02 | 58.21 | 43.67 | 33.55 | 48.83 | 62.72 | 29.78 | 19.73 | 32.55 | 48.46 | 61.93 | 48.59 | 70.48 | 80.84 |
| CyGNet | 20.22 | 12.35 | 21.66 | 35.82 | 37.65 | 27.43 | 42.63 | 57.90 | 40.42 | 29.44 | 46.06 | 61.60 | 27.12 | 17.21 | 30.97 | 46.85 | 62.98 | 50.97 | 70.60 | 80.98 |
| xERTE | 19.45 | 11.92 | 20.84 | 34.18 | 40.79 | 32.70 | 45.67 | 57.30 | 46.62 | 37.84 | 52.31 | 63.92 | 29.31 | 21.03 | 33.51 | 46.48 | 53.62 | 48.53 | 58.42 | 60.53 |
| RE-GCN | 19.69 | 12.46 | 20.93 | 33.81 | 42.00 | 31.63 | 47.20 | 61.65 | 48.03 | 37.33 | 53.90 | 68.51 | 32.62 | 22.39 | 36.79 | 52.68 | 62.30 | 48.53 | 59.27 | 78.58 |
| TITer | 18.19 | 11.52 | 19.20 | 31.00 | 41.73 | 32.74 | 46.46 | 58.44 | 47.60 | 38.29 | 52.74 | 64.86 | 29.98 | 22.05 | 33.46 | 44.83 | 61.28 | 49.35 | 68.30 | 79.77 |
| TiRGN | 21.67 | 13.63 | 23.27 | 37.60 | 43.81 | 33.49 | 48.90 | 63.50 | 49.84 | 39.07 | 55.75 | 70.11 | 33.58 | 23.10 | 37.90 | 54.20 | 62.95 | 50.34 | 69.37 | 80.92 |
| **TKG-LM** | **21.94** | **13.76** | **24.16** | **37.81** | **46.50** | **34.32** | **53.15** | **70.26** | **50.72** | **39.70** | **55.83** | **70.81** | **33.99** | **23.24** | **38.02** | **56.84** | **63.10** | **51.85** | **70.74** | **81.19** |

**Baselines.** The compared baselines include existing methods of multiple fields. (1) Static modeling methods: DistMult (Yang et al., 2014), ConvE (Dettmers et al., 2018), ComplEx (Trouillon et al., 2016), ConvTransE (Shang et al., 2019), RotatE (Sun et al., 2019). (2) SOTA Graph-based TKG methods:TTransE (Leblay & Chekol, 2018), RGCRN (Seo et al., 2018), RE-NET (Jin et al., 2019), CyGNet (Zhu et al., 2021), xERTE (Han et al., 2020), RE-GCN (Li et al., 2021), TITer (Sun et al., 2021), and TiRGN (Li et al., 2022). (3) Fine-tuned LLMs:T5 (Raffel et al., 2020), LLaMA (Zheng et al., 2023), ChatGLM (Zeng et al., 2022): 4) Graph+LM methods: Graph-Bert (Zhang et al., 2020), GraphToolFormer (Zhang, 2023). (5) KG+LM methods:StructGPT (Jiang et al., 2023), KG-BERT (Yao et al., 2019), KGS2S (Cheng et al., 2022).

**Implementation.** The architecture of the pre-trained LM in our TKG-LLM is the same in the step of scoring and modality fusion, defaults to Bert (Devlin et al., 2018) and can be extended to any multi-layer LMs such as RoBERTa (Li et al., 2021), GPT2 (Radford et al., 2019), etc. The number of layers of our GNN and Bert-Base is 3 and 12, and the hidden layer dimension $d$ is 200. As for the pruning and sampling prompt construction, the hop of neighbors $m$ is 2, the number of nodes on the pruned graph $\mathcal{G}_{sub}^{q}$ is 2000, and the number of sampling historical facts $k$ is 5. The two-loss terms $\alpha$ and $\beta$ are set to 0.7 and 0.1, respectively. The Adam optimizer (Kingma & Ba, 2014) and a multi-step learning rate scheduler are adopted for model training. The learning rate and the number of epochs are set to $1e^{-3}$ and 50. The evaluation metrics are the mean reciprocal rank (MRR) and hit rate (H@1, H@3 and H@10).

## 4.2 MAIN RESULTS

**Static and Temporal KG Baselines**. Table 1 reports the average performance of the TKG embedding baselines and the proposed TKG-LM on five standard TKG datasets. Most static modeling methods perform poorly as they neglect the critical and rich temporal information. Our TKG-LM achieves competitive results to state-of-the-art (SOTA) graph-based TKG methods. For example, on the ICEWS14 dataset, our method's MRR performance improves by 11.43% over TiTer, and 6.14% over the prior best graph-based TKG method, TiRGN. The boost confirms that TKG extrapolation tasks require the use of temporal information of facts and the semantic context of entities and relations.

**Fine-tuned LLMs**. As shown in Table 2, our TKG-LM shows consistent improvements over three popular large language models, including T5 (Raffel et al., 2020), LLaMa (Zheng et al., 2023), and ChatGLM (Zeng et al., 2022). These LLMs are fine-tuned using TKG-based prompt instructions that are manually constructed. Directly finetuning LLMs to TKG extrapolation scenarios generally results in sub-optimal performance because of the lack of fusion modules for extra multi-modal information. On the one hand, LMs are pre-trained on static corpora and less sensitive to time-aware questions. The same question at different times leads to conflicting facts. For example, the US President in 2016 could be both Barack Obama and Donald Trump. On the other hand, LMs

Table 2: Performance comparison (%) between LLMs, LM-enhanced methods and our TKG-LM using different LM encoders.

| Type | Dataset Method | ICEWS14 H@1 | ICEWS14 H@10 | YAGO H@1 | YAGO H@10 |
|------|------|------|------|------|------|
| LLMs | FastChat-T5-3B | 2.32 | 16.82 | 6.14 | 26.14 |
| | LLaMa-Vicuna-6B | 3.03 | 18.77 | 7.42 | 27.42 |
| | ChatGLM-6B-Int4 | 3.14 | 19.02 | 7.93 | 27.93 |
| Graph+LM | Graph-BERT | 13.77 | 37.56 | 24.75 | 61.10 |
| | GraphToolFormer | 11.98 | 33.75 | 18.17 | 53.75 |
| KG+LM | KG-BERT | 12.62 | 34.29 | 21.08 | 58.78 |
| | KG-S2S | 19.74 | 53.10 | 39.56 | 67.16 |
| | StructGPT | 26.45 | 55.90 | 44.91 | 72.47 |
| Ours | TKG-BERT-Base | 34.32 | 70.26 | 51.25 | 80.23 |
| | TKG-BERT-Large | **34.68** | **70.61** | **51.85** | **82.11** |
| | TKG-RoBERTa-Base | 34.44 | 70.96 | 51.70 | 81.19 |

Table 3: Ablation study on how MRR changes in various components of our TKG-LM (The last line) on ICEWS14 dataset.

| Scoring | Prompt | Fusion | Loss | MRR |
|---------|--------|--------|------|-----|
| w/o Purning | | | | 43.67(-2.83) |
| w/o $r_i$ | | | | 46.40(-0.10) |
| w/o $\mathcal{L}_{MLM}$ | | | | 44.75(-1.75) |
| | w/ Fixed | | | 46.05(-0.45) |
| | w/ Uni | | | 46.11(-0.39) |
| | w/ $\Delta_t$ | | | 46.22(-0.28) |
| | | w/ MLP | | 46.39(-0.11) |
| | | w/o Res | | 46.25(-0.25) |
| | | w/o $\psi$ | | 45.95(-0.55) |
| | | | w/o $\mathcal{L}_{lm}$ | 44.07(-2.43) |
| w/ g | w/ $\Phi$ | w/ Attn | w/ $\mathcal{L}$ | 46.50 (Ours) |

have a weak perception of spatial and topological factors, and they are unable to perform precise multi-step computational reasoning on history snapshots as GNNs can (Lee et al., 2023), which further degrades their performance on the extrapolation task.

**LM-Enhanced Models**. To further evaluate our method's performance, we compare our TKG-LM with the LM-Enhanced models, including Graph+LM and KG+LM. Our TKG-LM is superior to the methods of both fields on the ICEW14 dataset and YAGO dataset, as they neglect temporal information or edge relations, making them unsuitable for TKG extrapolation task. In addition, we compare the performance of different LM-encoder architectures, such as BERT-Base, BERT-Large, and RoBERTa-Base. Compared to BERT-Base, BERT-Large and RoBERTa-Base can bring more performance gains. As the number of parameters in the pre-trained LM increases, our method's performance improves, indicating that more prior knowledge is adequately utilized by our TKG-LM to obtain better results.

## 4.3 ABLATION STUDY

In this section, we analyze how various components of our TKG-LLM contribute to the final performance. Table 3 shows four parts of our method, including the scoring function (Scoring), the way of prompt construction (Prompt), the layer-wise fusion of modalities (Fusion), and the optimization object (Loss). w and w/o are short for with and without, respectively.

**Impact of the Gap Function.** Our TKG-LM adopts a time-weighted semantic gap $g$ in Equation 4 to score and prune historical facts. When irrelevant facts are kept (w/o Pruning), there will be a performance degradation of up to 2.83%. As more candidates usually mean more difficult learning problems, our pruning strategy is beneficial to reduce the search space for candidate entities and relations. When using the gap function for pruning, we ablate the two-term composition of the scoring function, i.e., only using the time-weighted ratio (w/o $\mathcal{L}_{MLM}$) or only using the semantic gap (w/o $r_i$). As shown in Table 3, their performance decreases by 1.75% and 0.1%, respectively, reflecting that semantics contributes much more to the results than the time difference. It demonstrates that pre-trained LMs' prior knowledge can adaptively capture valuable historical events based on diverse queries.

**Impact of the way of Prompt Construction.** When the language model uses the most recent historical events as input, with a fixed prompt instruction (w/ Fixed) for each query, its performance decreases by 0.45%. Regarding the sampling prompt, we compare the proposed distribution $\Phi$ of Equation 5 to two naive distributions: the uniform (w/ Uni) and exponential time difference (w/ $\Delta_t$) distributions. The latter works' performance is slightly better than the former because more recent times may be more likely to be important for the query. Also, our distribution works best because it incorporates both temporal and semantic context. Its additional information allows the training procedure to learn entity representations that simultaneously reflect facts from the knowledge base and associated history.

**Impact of the Modality Fusion Modules.** (Zhang et al., 2022) propose a fusion scheme that obtains the fusion embedding by concatenating two modality representations and translating them using an

Table 4: Top-5 ranking results of candidate entities for different methods. The first column includes the query for inference and their target entity.

| Query & Target | ChatGLM-6B-Int4 | RE-GCN | TKG-LM |
|---|---|---|---|
| (159, Barack Obama, Host a visit) → François Hollande | 5, Barack Obama, Xi Jinping, Angela Merkel, Benjamin Netanyahu, François Hollande | 4, Japan, South Korea, France, François Hollande, China | 1, François Hollande, France, Adbel fattah al sisi, China, Poland |
| (152, Activist Thailand , Make statement) → Thailand | 4, Activist Thailand, Military Thailand, Citizen Thailand, Thailand, iran | 3, National united front for democracy against dictatorship, Military Thailand, Thailand,Leader Thailand, Iron | 1, Thailand, National united front for democracy against dictatorship,Military thailand, Leader thailand, Citizen thailan, |

MLP (w/ MLP). According to Table 3, the performance of MLP is not as good as our attention fusion module. w/o Res represents no residual connection in Equation 14 and 15, and w/o $\psi$ means we do not use the time encoding function in Equation 10. Their performance is all sub-optimal because residual connections allow deeper layers to learn additional information on top of earlier layer features, capturing more abstract and complex patterns to improve model accuracy. The time-encoding function can Capture Temporal Patterns for the representation and modeling of temporal patterns in sequential data. They can encode chronological information, such as time steps or timestamps, into continuous representations that capture the relationship between different points in time.

## 4.4 CASE STUDY

In this section, we present several examples to show how our TKG-LLM can produce better predictions on TLG extrapolation tasks. Table 4 shows the Top-5 ranking results of candidate entities for two baselines, including the fined-tuned LLM ChatGLM, the classical TKG method RE-GCN, and our method. For the query $q$ =(Barack Obama, Host a visit, Nov 11), $q$'s subgraph visualization displays on Figure 1. On the one hand, the LM model ChatGLM tends to predict person names, such as Barack Obama, Xi Jinping, Angela Merkel, etc. One possible reason is that the text "host a visit" is usually semantically related to people. On the other hand, RE-GCN ranks Japan and South Korea at the top for the reason that Japan and South Korea have the most interactions with Obama. Due to the lack of pruning, the message propagation of RE-GCN will naturally be biased towards high-frequency neighbors, thus misleading the prediction.

As for our TKG-LM, it outperforms the other two methods on three different query examples and makes correct and accurate ranking predictions From the visualization of $q$'s subgraph in Figure 1 and the events' scores in Appendix B.1, it can be seen that both France and Hollande have similar interaction histories with Obama. The GNN layer of TKG-LM tends to predict entities with similar topological structures, which partly explains why the first two predictions of our method are Hollande and France. In addition, the LM -layer of TKG-LM can identify the semantic relationship between France and Hollande, that is, after Obama visits France, it is very likely to host a visit with Hollande. The prior knowledge and reasoning ability of LMs helps the model find the final answer.

## 5 CONCLUSION AND FUTURE WORK

In the paper, we introduce TKG-LM, a new model that incorporate textual data into temporal knowledge graph embeddings. We utilize the language model's prior knowledge to filter out irrelevant events, thereby decreasing the search space for answers. To improve the robustness of LMs on TKG extrapolation prediction, we construct adaptive and variable sampling prompt instructions as the input of LMs. Our method integrates the encoded representation of LMs and graph neural networks in a multi-layer attention-based module to enable bidirectional information flow between two modalities. In both TKG extrapolation and LM-enhanced domains, our method outperforms state-of-the-art(SOTA) TKG embedding methods, classical fine-tuned LLMs, existing Graph+LM methods and KG+LM models on various datasets. Extensive experiments on ablation demonstrate that our TKG-LLM considerably improves the TKG extrapolation performance.

One limitation is that our method has a higher time and space complexity. In the future, we will explore work on a wider range of knowledge graphs combined with large language models.

ACKNOWLEDGMENTS

Use unnumbered third level headings for the acknowledgments. All acknowledgments, including those to funding agencies, go at the end of the paper.

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

# A  EXPERIMENTAL SETUP

## A.1  DATASET

We use five public TKG datasets to evaluate the effectiveness of the proposed TKG-LM. They are GDELT (Leetaru & Schrodt, 2013), ICEWS14 (Trivedi et al., 2017), ICEWS05-15 (García-Durán et al., 2018), ICEWS18 (Boschee et al., 2015) and YAGO (Mahdisoltani et al., 2013). ICEWS14, ICEWS05-15, and ICEWS18 datasets, is from the Integrated Crisis Early Warning System. The YAGO is supplemented with time information based on the traditional static KGs YAGO3. GDELT is from the Global Database of Events, Language, and Tone. The statistics of four TKG datasets are summarized in Table 5.

## A.2  BASELINES

The static KG reasoning models compared with our work are shown as follows: DisMult (Yang et al., 2014), a model that proposes a simplified bilinear formulation to capture relational semantics. ConvE (Dettmers et al., 2018), a model that adopts a 2D convolutional neural network to model the interactions between entities and relations. ComplEx (Trouillon et al., 2016), a model that converts the embedding into complex vector space to handle symmetric and antisymmetric relations. RotatE (Sun et al., 2019), a model that defines each relation as a rotation from the subject entity to the object entity in the complex vector space.

Table 5: Detailed Information about the five involved datasets.

| Datasets | #Entities | #Relations | Time Interal | #Train | #Valid | #Test |
|---|---|---|---|---|---|---|
| GDELT | 7691 | 240 | 1 Day | 1734399 | 238765 | 305241 |
| ICEWS14 | 7128 | 230 | 1 Day | 74845 | 8514 | 7371 |
| ICEWS05-15 | 10488 | 251 | 1 Year | 368868 | 46302 | 46159 |
| ICEWS18 | 23033 | 256 | 1 Day | 373018 | 45995 | 49545 |
| YAGO | 10623 | 10 | 1 Year | 161540 | 19523 | 20026 |

We compare e compare the performance of our proposed TKG-LM model with that of multiple static and dynamic modeling methods (including interpolation and extrapolation). Note that the static methods are trained without 7 the time dimension and the interpolation methods are trained with both historical and future data; thus, they are not good at future event forecasting when provided with only historical

## A.3  IMPLEMENTATION

The parameters of model architectures are fixed: all methods' structural encoder layers are 2, and the hidden layer dimension is 128. For a fair comparison, the downstream classifier for all methods is a trainable two-layer perceptron. The Adam optimizer (Kingma & Ba, 2014) and an early-stopping strategy are adopted for model training. Except for the above data augmentations, the two students and HWM also accept random horizontal flipping, Cutout (?), and Random Augment (?), respectively. The SGD optimizer (?) is adopted with a learning rate of $0.1$ and a weight decay of $5e^{-4}$. The number of epochs and the batch size are set to 300 and 128, respectively. For the settings of ImageNet, we employ the standard ResNet18 (?) as the backbone and train 100 epochs with a learning rate of $0.1$.

The evaluation metrics use the mean reciprocal rank mrr and hit rate, since prediction usually involves ranking the scores of missing graph elements at future times. For our encoder, we use the exact same architecture as GreaseLM (19 LM layers followed by 5 text-KG fusion layers; 360M parameters in total). We initialize parameters in the LM component with the RoBERTa-Large release and initialize the KG node embeddings with pre-computed ConceptNet entity embeddings. For the link prediction objective, we use DistMult for KG triplet scoring, with a negative exampling of 128 triplets and a margin of.

# B  DETAILS

## B.1  EXAMPLES

We show two kinds of queries with their corresponding historical events and scoring function values.

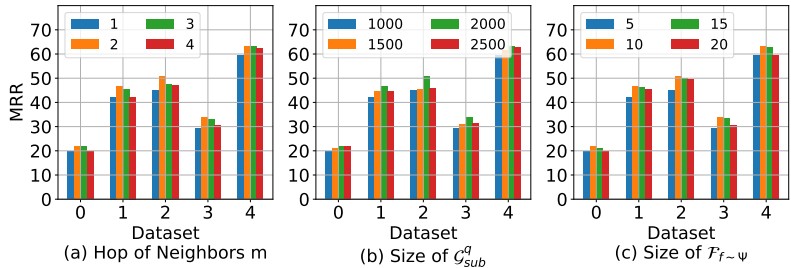

Figure 2: Enter Caption

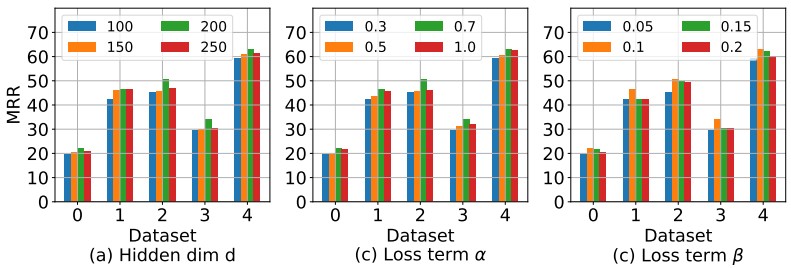

Figure 3: Enter Caption

Table 6: Example for the query, part of historical facts and the corresponding gap.

| | | $g(f \mid q, LM)$ |
|---|---|---|
| **Query** $q$ | 159, Barack Obama, Host a visit, ? | |
| **Historical Facts** | 156, barack obama, express intent to meet or negotiate, abdel fattah al sisi, 9.97 | 9.97 |
| $f \in F_{sub}^q$ | 156, françois hollande, express intent to meet or negotiate, barack obama, 10.08 | 10.08 |
| | 156, france, make a visit inversely, barack obama, 10.39 | 10.39 |
| | 156, france, host a visit, barack obama, 10.48 | 10.48 |
| | 155, barack obama, express intent to meet or negotiate, france, 13.64 | 13.64 |
| | 155, françois hollande, express intent to meet or negotiate, barack obama, 13.79 | 13.79 |
| | 155, barack obama, express intent to meet or negotiate, françois hollande, 14.06 | 14.06 |
| | 155, barack obama, criticize or denounce, france, 14.18 | 14.18 |
| | 155, barack obama, make an appeal or request, china, 14.2 | 14.2 |
| | 155, barack obama, engage in negotiation, france, 14.46 | 14.46 |
| | 155, barack obama, make an appeal or request, china, 14.47 | 14.47 |
| | 155, barack obama, express intent to meet or negotiate, france, 14.57 | 14.57 |
| | 155, barack obama, make statement, france, 14.67 | 14.67 |
| | 156, barack obama, make a visit france, 14.71 | 14.71 |
| | 156, barack obama, consult, françois hollande, 14.96 | 14.96 |
| | 154, barack obama, express intent to meet or negotiate, abdel fattah al sisi, 15.01 | 15.01 |
| | 155, barack obama, make pessimistic comment, france, 15.15 | 15.15 |
| | 154, françois hollande, express intent to meet or negotiate, barack obama | 15.19 |
| | 154, barack obama, express intent to provide economic aid, poland | 15.35 |
| **Query** $q$ | 152 activist thailand make statement ? | |
| **Historical Facts** | 149, activist thailand refuse to de escalate military engagement inversely thailand | 14.56 |
| | 93, activist thailand make an appeal or request inversely national united front for democracy against dictatorship | 14.60 |
| | 149, activist thailand refuse to de escalate military engagement military thailand | 14.81 |
| | 149, activist thailand criticize or denounce inversely activist thailand | 14.91 |
| | 148, activist thailand praise or endorse coup d etat leader thailand | 15.05 |
| | 9, activist thailand appeal for diplomatic cooperation such as policy support citizen thailand | 15.13 |
| | 148, activist thailand use tactics of violent repression inversely coup d etat leader thailand | 15.13 |

