# OpenReview forum: "TKG-LM: Temporal Knowledge Graph Extrapolation Enhanced by Language Models"
_ICLR.cc/2024/Conference — ICLR 2024 Conference Withdrawn Submission_

### Official Review · Reviewer_SrU9 · 2023-10-30

**Soundness:** 2 fair
**Presentation:** 2 fair
**Contribution:** 2 fair
**Rating:** 3
**Confidence:** 4

**Summary:**

The paper focused on the task of temporal knowledge graph (facts) extrapolation (prediction). The main contribution is to make historical facts selection by employing LMs. The selected historical facts are combined with textual information and LMs to predict time-related facts more precisely. The main contributions include 1) Exploiting an LM-based method for historical facts selection; 2) devising an adaptive prompt and pruned subgraph for temporal reasoning; 3) Proposing a layer-wise modality interaction with an attention-based residual fusion module. The authors also conduct experimental comparisons on five open-shared datasets. The results show that the proposed methods could obtain SOTA performance than existing TKG methods, LLM, and LM+KG models.

**Strengths:**

1) Employing the knowledge in LMs for the TKG task is straightforward and interesting.
2) Capturing the interaction between knowledge from LMs and existing KGs is somewhat novel, although this operation could be seen in previous knowledge-enhanced models for other NLP tasks, including text classification, question answering, etc.
3) The comparisons in the experimental part are sound. Several different kinds of baselines are selected which makes the results more convincing.

**Weaknesses:**

1) The whole paper is not very clear. In the introduction part, the authors should show some examples to illustrate what is the task of TKG, and what are difficulties of the three claimed weaknesses (adequate utilization of the semantic prior knowledge of LMs, robust temporal reasoning, and effective interaction of multimodel information).

2) Actually, I don't like the term multimodel for knowledge and texts. Knowledge could not be regarded as a kind of modality.

3) In the introduction part, what the is behind reasons about "the sufficient structured semantics of TKG can hardly be represented by semantic-agnostic GNNs".  The authors should explain it more clearly.

4) The authors should give out the experimental results of fine-tuned LLMs and Graph/KG enhanced LM models for the other three datasets.

**Questions:**

1) I wonder if unseen facts could be predicted based on many existing facts following the time dimension. For example, if we never know the president of the USA in 2020, how could we predict it according to all the presidential information of USA in the history?

2) In equation (7), why the contexts should be [t,s,r] rather than [s,r,t] or [s,t,r]?

3) How about the performance goes when we use LLMs with different parameter sizes, such as LLaMa-13b or 65b? How about the performance of ChatGPT or GPT-4?

4) Where are the texts come from in the equation (3)?

---

### Official Review · Reviewer_CAix · 2023-10-31

**Soundness:** 3 good
**Presentation:** 3 good
**Contribution:** 2 fair
**Rating:** 5
**Confidence:** 4

**Summary:**

This paper presents a temporal knowledge reasoning model enhanced by the LM models.  The authors conduct extensive experiments on five benchmark datasets. The results show the effectiveness of the proposed method. The paper is well written and the solution is clear.

**Strengths:**

1. The authors conduct extensive experiments on five benchmark datasets. The results show the effectiveness of the proposed method.
2. The paper is well written and the solution is clear.

**Weaknesses:**

1. The motivation is not well established. In some cases, the entities do not have any text description especially for the new entities. In this case, how the LM benefits the KG reasoning.
2. In Table 1, many strong baselines published in recent years are not considered, making the contribution of this paper is not significant.
Xu et al., 2023. Temporal Knowledge Graph Reasoning with Historical Contrastive Learning.
Zhang et al., 2023. Learning Long- and Short-term Representations for Temporal Knowledge Graph Reasoning.

**Questions:**

1. The motivation is not well established. In some cases, the entities do not have any text description especially for the new entities. In this case, how the LM benefits the KG reasoning.
2. In Table 1, many strong baselines published in recent years are not considered, making the contribution of this paper is not significant.
Xu et al., 2023. Temporal Knowledge Graph Reasoning with Historical Contrastive Learning.
Zhang et al., 2023. Learning Long- and Short-term Representations for Temporal Knowledge Graph Reasoning.

---

### Official Review · Reviewer_Xuok · 2023-11-03

**Soundness:** 3 good
**Presentation:** 4 excellent
**Contribution:** 2 fair
**Rating:** 5
**Confidence:** 4

**Summary:**

The author proposed TKG-LM to fuse GNN-based knowledge graph embedding method and large language models. The model leveraged LM to create subgraphs and encode temporal semantic information, while using a multi-modality residual learning framework to incorporate the learned embeddings from GNN and LM.

**Strengths:**

S1. The overall writing is sound with a clear presentation.

S2. The author conducted extensive experiments and showed promising results. Although the topic has been previously explored, the method is fairly novel and the author apparently approached the problem from a different perspective.

S3. One challenge of the previous paper is that it failed to leverage any neighborhood information. By using the multi-modal learning, the author addressed this problem.

S4. It is also worth noting that the paper introduces a new general direction on combining GNN and LM. Previous works mostly leverage the LM in a heuristic way by applying descriptive graph languages on LM. This work minimized the heuristic part and was able to optimize the strategy through the joint optimization.

**Weaknesses:**

W1. The author needs to re-visit their claim regarding the novelty of the paper and acknowledge the latest research in the field. For example, it’s missing "Pretrained Language Model with Prompts for Temporal Knowledge Graph Completion" (https://arxiv.org/abs/2305.07912), and "Graph Neural Prompting with Large Language Models" (https://arxiv.org/abs/2309.15427).

W2. The author needs to include the first paper above as an additional baseline. It first leverages the power of large language model to tackle the TKGC task by directly input the text sequence generated from the knowledge graph.

Also note the following possible typo on Page 4 in the second paragraph - "make fully use"--> "make full use"

**Questions:**

Q1. What are the comparison results in relation to W2 above?

Q2. How would the authors revise their novelty in light of the missed related work?

---

### Official Review · Reviewer_ZuLH · 2023-11-04

**Soundness:** 3 good
**Presentation:** 1 poor
**Contribution:** 2 fair
**Rating:** 3
**Confidence:** 4

**Summary:**

The paper introduces a methodology that combines graph neural networks with language models to address the temporal knowledge graph (TKG) extrapolation challenge.
Initially, the approach utilizes the prior knowledge within language models to narrow down the search space by eliminating irrelevant events.
Subsequently, it formulates dynamic prompt instructions to fine-tune the language models.
Following this, the method integrates the derived representations from language models with those from graph neural networks using a layered attention mechanism to yield the final predictions.

The framework presented surpasses other established graph-based methods for temporal knowledge graph (TKG) tasks in its performance.
The paper claims superiority over several other research directions, such as standalone large language models (LLMs), combined Graph and LM approaches, and knowledge graph-enriched language models (KG-LM).
However, the paper's argument is less persuasive due to the insufficient details provided, particularly regarding the LLMs.

**Strengths:**

The presented framework offers an innovative method of merging the prior knowledge from bidirectional pre-trained models (such as BERT and RoBERTa) with graph neural networks to enhance the accuracy of extrapolation predictions.
This proposed system consistently surpasses traditional graph-based methods in performance.

**Weaknesses:**

Despite of interesting results, I have following minor concerns.

**1. Lack of technical details**
The paper does not clearly explain the training process for the baseline models, including the specifics of the training prompts used.

**2. Lack of baselines.**
The paper points out potential issues with using fixed prompt templates, namely overfitting and reduced model generalization, citing the TKG-ICL [1] study for support.
Nonetheless, it doesn't provide a head-to-head comparison with TKG-ICL's approach.
Additionally, the concept of "prompts" might be interpreted differently here compared to their use in the TKG-ICL study.
This research uses encoder-only transformers trained on a TKG dataset, in contrast to TKG-ICL's reliance on decoder-only transformers for zero-shot inference without TKG dataset training.
Additionally, it appears that the "prompts" as described in this paper are essentially "various combinations of demonstration examples" rather than the format and style typically associated with prompts.
Consequently, while it may be reasonable to assume that TKG-ICL isn't designed for multi-hop reasoning tasks, the critique in this paper would be more persuasive if it included a direct comparison with the TKG-ICL methodology.

*[1] Temporal Knowledge Graph Forecasting Without Knowledge Using In-Context Learning., Lee et al., 2023.*

**Questions:**

**Q1**. What are the TKG-based prompt instructions for LLM training ? Seems the results on LLMs (FastChat, LLAMA-Vicuna, ChatGLM) are not convincing since there is a recent work [1] showing similar approach shows much better results than reported numbers in this paper.

*[1] GenTKG: Generative Forecasting on Temporal Knowledge Graph, Liao et al, 2023*

**Q2**. Are there instances where the TKG-LM methodology demonstrates proficient performance in multi-hop reasoning, thereby reinforcing the argument `Manually constructing topology-relevant prompt instructions will cause LMs to over-rely on simple, or even spurious, patterns to find shortcuts to answers, leading to overfitting and reducing generalization`?

[Grammar & Writing]
- Missing citations in Appendix